# Understanding the Resource Cost of Fully Homomorphic Encryption in Quantum Federated Learning

## Abstract

Quantum Federated Learning (QFL) enables distributed training of Quantum Machine Learning (QML) models by sharing model gradients instead of raw data. However, these gradients can still expose sensitive user information. To enhance privacy, homomorphic encryption of parameters has been proposed as a solution in QFL and related frameworks.

In this work, we evaluate the overhead introduced by Fully Homomorphic Encryption (FHE) in QFL setups and assess its feasibility for real-world applications. We implemented various QML models including a Quantum Convolutional Neural Network (QCNN) trained in a federated environment with parameters encrypted using the CKKS scheme. This work marks the first QCNN trained in a federated setting with CKKS-encrypted parameters. Models of varying architectures were trained to predict brain tumors from MRI scans. The experiments reveal that memory and communication overhead remain substantial, making FHE challenging to deploy. Minimizing overhead requires reducing the number of model parameters, which, however, leads to a decline in classification performance, introducing a trade-off between privacy and model complexity.

## 1 Introduction

Federated Learning (FL) (McMahan et al., 2017) offers an effective way to train machine learning models in a distributed way. It is an iterative protocol in which multiple clients train local models using their own private datasets. Instead of sharing raw data, clients transmit the corresponding gradients to a central party, which then aggregates them to update a global model. FL is particularly advantageous when transmitting data to a central party is impractical or maintaining a centralized dataset is too costly. Moreover, FL plays a crucial role in ensuring compliance with data protection regulations, such as the Personal Data Protection Act (PDPA) (Parliament of Singapore, 2012), the General Data Protection Regulation (GDPR) (European Parliament, 2016) and the Digital Personal Data Protection Act (DPDPA-2023) (Parliament of India, 2023).

With advancements in quantum computing, adapting classical algorithms to quantum environments is becoming increasingly relevant, including in FL. Quantum methods may offer significant speedups in a variety of machine learning tasks compared to conventional ML techniques (Li & Deng, 2025). Special attention has been paid by Chehimi & Saad (2022) to demonstrate the feasibility of enabling FL workflows in a quantum environment using TensorFlow Quantum (TFQ) (Broughton et al., 2021). However, this approach relies on traditional 5G cellular infrastructure for parameter sharing, which poses risks of exposing sensitive information.

While FL circumvents direct data sharing, the weight parameters of local models can still potentially reveal sensitive information. In a classical federated learning setup, a semi-honest central party could exploit these weights to extract specific data points, e.g., by leveraging *Deep Leakage from Gradient* (Zhu et al., 2019). Additionally, it might try to determine whether a specific sample was used in the training process (membership inference attacks) (Shokri et al., 2017), or obtain a sample from each class (model inversion attacks) (Fredrikson et al., 2015) if it has access to the final or an intermediate model. Since Quantum Federated Learning (QFL) frameworks are often hybrid, with parameters transmitted over existing wireless networks, it is crucial to secure these parameters before transmission (Chehimi & Saad, 2022).

There are different ways to protect clients against untrustworthy central parties. One such option is to leverage Differential Privacy (DP) (Abadi et al., 2016; Pathak et al., 2010). DP aims to reduce the information that model weights reveal about individual data points by adding noise or clipping gradients, making it generally suitable for cross-device settings (Abadi et al., 2016). However, in cross-silo setups—where each client, such as a hospital, holds highly sensitive data—DP might not satisfy the necessary privacy requirements, since it discloses plain text gradients that can be exploited to extract user information (Zhang et al., 2020; Phong et al., 2018). Another approach is Secure Multi-Party Computation (SMPC) (Goldreich, 1998), which prevents inference attacks by enabling participants to collaboratively compute a function while maintaining the confidentiality of their inputs. SMPC is particularly advantageous in FL for securely aggregating gradients. Nonetheless, multiple parties might collude and compromise this guarantee (Pati et al., 2024). Such collusion is not possible when Fully Homomorphic Encryption (FHE) (Gentry, 2009) is employed. FHE refers to encryption schemes that allow additions and multiplications to be performed directly on encrypted data, such that decrypting the results yields the same value as performing the corresponding algebraic operations on the original plain text. FHE, which is becoming an industry standard (Albrecht et al., 2018), offers substantial privacy guarantees but introduces significant computational overhead and requires careful management of public and private keys. Pati et al. (2024) posit that FHE inherently provides a robust solution to ensure privacy in federated learning. Notably, QFL encodes data as quantum states and may achieve the same accuracy with fewer parameters than traditional FL (Chen & Yoo, 2021), potentially making FHE more feasible since fewer parameters need to be encrypted.

This work provides a robust implementation of FHE within a QFL setup, building upon the foundation laid by Dutta et al. (2024). While their study employed simpler Quantum Machine Learning (QML) models, this study implements a variety of algorithms: models without quantum layers, with simple quantum layers, and with a Quantum Convolutional Neural Network (QCNN) (Cong et al., 2019). This work is, to our knowledge, the first to provide an implementation of a QCNN within FL that incorporates FHE for parameter encryption. For feature extraction, two variations are tested: a self-trained Convolutional Neural Network (CNN) and a pre-trained ResNet-18 (He et al., 2016).

Secondly, this work analyzes the overhead introduced by FHE. In addition to measuring training times, CPU and memory utilization and communication overhead were assessed in an experiment predicting tumors using MRI scans (Nickparvar, 2021). These insights help determine whether FHE can realistically be applied in real-world scenarios and identify the computational and resource requirements involved. The goal is to answer whether FHE for parameter encryption is merely a theoretical concept or suitable for production use.

The analysis revealed that integrating FHE into QFL results in substantial memory and communication overhead, rendering it largely impractical for real-world applications. To mitigate this overhead, the number of parameters must be reduced, which in turn leads to diminished classification performance. Hence, introducing FHE into QFL comes with a trade-off between privacy and model complexity. While transfer learning—offering complex models with a low parameter count—shows promise, this work demonstrates that it is only a partial solution to this trade-off.

## 2 Related Works

Integrating homomorphic encryption techniques into conventional FL setups has gained significant attention in recent years, leading to numerous works (Qiu et al., 2022; Yao et al., 2023; Hijazi et al., 2024; Fang & Qian, 2021; Jin et al., 2024; Pan et al., 2024; Hannemann & Buchmann, 2023). For example, FedML-HE (Jin et al., 2024) provides an FL framework that selectively encrypts only the most sensitive gradients using CKKS (Cheon et al., 2017). This approach aims to reduce communication and computational overhead while maintaining the desired level of privacy. The authors demonstrated that encrypting the top 30% most sensitive parameters, along with the first and last layers, effectively mitigates inversion attacks. A very effective approach to employing CKKS in a FL setup was proposed by Jiang & Ju (2022), though their method relies on hardware acceleration and advanced, GPU-based cryptographic implementations to mitigate the overhead of homomorphic encryption. In contrast, the present work focuses on a general, hardware-agnostic assessment of the resource requirements for FHE in FL and QFL environments. This allows the results to be

interpreted independently of specialized hardware optimizations, but also implies that observed runtimes and memory usage are likely higher than what could be achieved using hardware-accelerated implementations.

Several methods have also been proposed to secure user data in QFL (Chen & Yoo, 2021; Innan et al., 2024) setups. Examples include Quantum Secure Aggregation (Zhang et al., 2022), which secures parameters by encoding them into qubits and transmitting them via quantum channels, and CryptoQFL (Chu et al., 2023), which uses Quantum One-Time Pads (QOTP) to encode quantum states. In CryptoQFL, the QOTP keys are homomorphically encrypted before transmission to the central party, offering a dual-layer security approach. Additionally, it is feasible to directly apply homomorphic encryption to user data within a quantum delegated framework (Li & Deng, 2025).

However, the use of FHE to directly encrypt parameters in QFL setups remains largely unexplored. The most relevant work in this area is presented by Dutta et al. (2024), who provide a practical implementation of a QFL framework integrating CKKS encryption. They evaluated four different models: a conventional CNN and a CNN combined with simple quantum layers, each tested both with and without FHE. The models were assessed on multiple datasets, including CIFAR-10 (Krizhevsky et al., 2009), Brain MRI (Nickparvar, 2021), and PCOS (Hub et al., 2024). While the use of FHE resulted in longer training times, test accuracy was minimally affected. For example, the QNN model trained on the Brain MRI dataset achieved an accuracy of 88.75% with FHE compared to 89.71% without FHE. Training time increased from 110.6 minutes to 116.5 minutes when employing FHE. However, it is worth noting that the source code indicates not all layers of the model were encrypted. Building on their previous work, the authors also proposed a multimodal quantum mixture of experts (MQMoE) model integrated with FHE (Dutta et al., 2025). This method enables processing of multiple data sources to extract features for classification.

In a nutshell, the integration of FHE into FL frameworks has been widely explored, but connections to models leveraging quantum capabilities remain underdeveloped. Although the computational and communication overhead introduced by FHE is well known, existing studies primarily focus on training time and often lack detailed, accurate statistics (Pan et al., 2024; Hijazi et al., 2024; Dutta et al., 2024; 2025). Such statistics are limited because they heavily depend on specific hardware and network configurations. To address this, our work provides a thorough and transparent analysis of FHE overhead in both FL and QFL setups. By directly measuring overhead and various time-related metrics under clearly defined conditions, we offer reproducible insights that complement existing studies and help contextualize the practical impact of FHE across different computational paradigms. This enables industry projects to better assess whether applying FHE to their specific use cases is beneficial.

## 3 Background

This work simulates an environment where multiple clients collaboratively train QML models with the aid of a central party (Swaminathan & Akgün, 2025). Following the FL protocol, the central party initializes gradients before the first training round. After receiving these gradients, each client trains a local model combining conventional and quantum layers. Before passing the updated gradients back to the central party, they are homomorphically encrypted. It is important to note that the central party only has access to the public key, whereas clients share the secret key. This approach prevents the central party, as well as any intermediary attempting to intercept the communicated data, from accessing plaintext gradients. The central party is then responsible for aggregating the weights and distributing them again among the clients. This process is typically repeated for multiple rounds. Figure 1 visualizes this setup.

### 3.1 Variational Quantum Circuits

Within a QFL environment, clients have access to quantum computing resources, enabling them to collaboratively train models that combine classical and quantum layers. This is achieved through training Variational Quantum Circuits (VQCs) (Chen & Yoo, 2021). VQCs, also referred to as Quantum Neural Networks (QNNs), leverage qubit rotations to parametrize quantum circuits. The corresponding parameters control the extent of rotation, enabling a learnable unit block whose parameters can be trained similarly to weights in conventional neural networks (Gurung et al., 2023). The general idea of VQCs is that a quantum

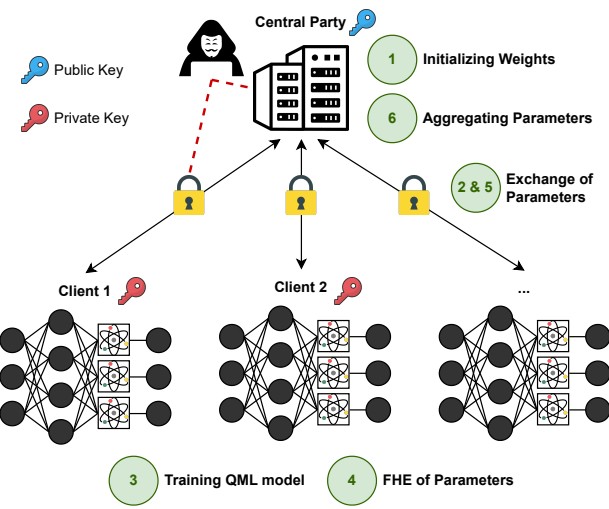

Figure 1: Overview of a QFL setup utilizing FHE for parameter encryption.

routine $E(x)$ transforms classical data $x$ into quantum states. Subsequently, a quantum circuit $W(\phi)$, which depends on a set of parameters $\phi$, is applied. In our simulations, these parameters are iteratively updated using conventional machine learning methods on a classical computer. Finally, the output of the quantum circuit is obtained by performing measurements (Chen & Yoo, 2021).

A crucial component of this process is embedding classical data into qubits. To accomplish this, it is common to reduce the dimensionality of the input data to align with the number of available qubits. Subsequently, a *variational encoding* scheme is employed, utilizing input values as rotation angles for single-qubit rotation gates. Given typical constraints on circuit depth and the number of qubits, hybrid learning schemes are often adopted to embed data into qubits. For example, a (pretrained) CNN can be employed to convert image data into a compact feature vector with significantly reduced dimensionality. The length of this feature vector is designed to match the number of qubits. Using the *variational encoding* scheme, this vector is then transformed into qubit states (Chen & Yoo, 2021).

Alongside simpler VQCs, this work also leverages QCNNs, which are a specialized type of VQCs inspired by classical convolutional neural networks. QCNNs inherit a similar structure, utilizing convolutional and pooling layers before making predictions via qubit measurements.

## 3.2 CKKS

To secure the trained parameters shared with the central party, we employ CKKS (Cheon et al., 2017) to homomorphically encrypt gradients. CKKS supports fast, approximate homomorphic computations on encrypted data, enabling arithmetic operations that yield results close to the true values. This approach is practical because, in many real-world applications, obtaining approximately correct results is sufficient.

Assuming a message $m$ is encrypted, the encryption and decryption process begins by scaling the input message $m$ with a global scale factor $\Delta$, which manages precision. Due to rounding of the scaled value, this introduces a small approximation error. A typical value for $\Delta$ is $2^{40}$. Since CKKS is based on the Ring Learning with Errors (RLWE) problem, the plaintext message must be encrypted using a secret key $sk$ into a polynomial in the ring

$$\mathcal{R}_Q := \mathcal{R}/Q\mathcal{R} \quad \text{with} \quad \mathcal{R} := \mathbb{Z}[X]/(X^N + 1). \tag{1}$$

Here, $N$ is a power of two and determines the polynomial degree. This parameter $N$ plays a key role in managing precision and capacity: a larger $N$ allows encoding of more data slots or larger numbers, which

reduces information loss and improves accuracy. The ring $\mathcal{R}_Q$ consists of polynomials whose coefficients are reduced modulo the integer $Q$. Importantly, $Q$ is not a single large modulus but a product of multiple moduli, often referred to as levels:

$$Q = q_0 \times q_1 \times \cdots \times q_i, \quad i \in \mathbb{N}_0. \tag{2}$$

Alongside the global scale $\Delta$ and the polynomial degree $N$, the sizes and structure of these moduli $q_i$ are crucial parameters for CKKS, as they directly influence precision, noise growth, and the number of homomorphic operations that can be securely performed. Other important concepts in CKKS include rescaling, which manages the multiplicatively growing global scale $\Delta$ and noise level, as well as relinearization, which mitigates the increasing size (or degree) of ciphertexts resulting from multiplications.

## 4 Experimental Setup

The framework was implemented in Python, leveraging TenSEAL (Benaissa et al., 2021) for homomorphic encryption, Flower (Beutel et al., 2020) for federated learning, and Pennylane (Bergholm et al., 2022) for quantum simulation. Gradients were aggregated using the FedAvg algorithm McMahan et al. (2017).

Generally, six different model types were analyzed that could be applied to a variety of real-world scenarios:

- `cnn`: A self-trained CNN without any quantum layers.

- `cnn-qnn`: A self-trained CNN followed by 6 Basic Entangled Layers (Idan & Jayakody, 2023) tailored for 4 qubits.

- `cnn-qcnn`: A self-trained CNN followed by a QCNN using 8 qubits.

- `resnet18`: A pre-trained ResNet-18 without any quantum layers.

- `resnet18-qnn`: A pre-trained ResNet-18 followed by 6 Basic Entangled Layers tailored for 4 qubits.

- `resnet18-qcnn`: A pre-trained ResNet-18 followed by a QCNN using 8 qubits.

In the following, model names are prefixed with *FHE* (e.g., `FHE-cnn`) when the experiment was performed with FHE. In contrast, the prefix *Standard* (e.g., `Standard-cnn`) indicates that no FHE was employed. The models have a similar number of parameters, ranging between 2,052 and 2,241, except for the `resnet18-qcnn` model, which has 4,145 parameters due to additional parameters required to combine the pre-trained model with the QCNN.

When quantum layers were integrated, most parameters were used for feature extraction. For example, the actual QCNN model requires only 21 parameters; the remainder corresponds to a conventional model that reduces dimensionality to enable embedding image data into qubits. The idea was, therefore, to use a pre-trained ResNet-18 model for feature extraction instead of training a separate CNN from scratch. In this case, only one layer needs to be tuned and sent over the network. Thus, combining ResNet-18 with quantum layers enabled the use of a complex model for feature extraction while keeping the number of parameters low to facilitate the use of FHE. This approach is not new; a similar strategy was implemented by Chen & Yoo (2021) in their *Hybrid Quantum-Classical Transfer Learning* models.

The analysis was conducted ten times for each model type—five runs utilizing FHE and five without—resulting in a total of 60 runs. All experiments simulated a setup consisting of one central party and 20 clients. However, due to resource constraints, the experiment involving the ResNet-18 combined with FHE and a QCNN was limited to a single client. Since this model has twice as many parameters, it required significantly more resources for encryption. Clients used a batch size of 32 and a learning rate of 0.001 to train the model for 20 rounds with 10 epochs each. All experiments were conducted on a high-performance computing cluster running Rocky Linux 8. Each evaluation was executed on a single compute node, requesting 48 logical CPU cores of type AMD EPYC 7551P (2.0–3.0 GHz) and 200 GB of ECC DDR4 memory. This configuration provided sufficient computational and memory capacity to support the simulation of QFL.

### 4.1 Metrics

During the experiments, metrics captured both static properties—such as the number of trainable parameters—and dynamic measurements recorded on both the central party side and the client side.

In detail, the analysis included training times such as the total training time, the central party round time (the time taken to complete one round of training on the central party side, measured from when parameters are sent to clients until all client updates are received), and the client round time (the time for a client to perform one round of training). Furthermore, parameter aggregation time as well as encryption and decryption time were tracked.

To assess computational overhead, virtual and real RAM usage was reported at intervals of 0.5 seconds for client and central party processes. The same applies for CPU usage, which was reported as a percentage of one core at intervals of 0.5 seconds.

Classification performance was measured by assessing accuracy, loss, recall, precision, and F1-score. Clients computed all of these metrics using a local test dataset and submitted the results to the central party, which then aggregated them. If parameters were not encrypted, the central party was able to use a centralized test set to compute centralized values for accuracy and loss. To distinguish between these statistics, we refer to them as aggregated accuracy and central accuracy, respectively.

Lastly, communication overhead was tracked by measuring the byte size of the parameters sent and received by the central party during one round of training.

### 4.2 Dataset

The experiments are conducted using the Brain Tumor MRI dataset (Nickparvar, 2021), as it is a common benchmark dataset and the medical domain is privacy-sensitive. It contains a total of 7,023 magnetic resonance imaging (MRI) scans, each categorized into one of four classes: glioma, meningioma, pituitary tumor or no tumor. The dataset comes pre-classified into training and test sets. The corresponding distribution across classes is shown in Table 1.

As mentioned in 4.1, the designated test dataset is assigned exclusively to the central party. If parameters are not encrypted, the central party can use these samples to compute accuracy and loss. The remaining images—listed under the *Train* column in Table 1—are distributed among the clients for local training and validation. After applying a train-validation split of 90/10, each client trains on approximately 257 samples and validates on 28. Before feeding the images into the models, input data is normalized and resized to 224×224 pixels.

Table 1: Class distribution of MRI dataset

| Class | Train | Test |
|---|---|---|
| glioma | 1321 | 300 |
| meningioma | 1339 | 306 |
| pituitary | 1457 | 300 |
| no tumor | 1595 | 406 |

## 5 Results

### 5.1 Increase of Training and Parameter Aggregation Times

Intuitively, time-related metrics are commonly used to measure overhead (Pan et al., 2024; Hijazi et al., 2024; Dutta et al., 2024; 2025). It becomes clear that employing FHE to encrypt roughly 2,000 parameters reliably increases training time as shown in Table 2. For instance, `Standard-cnn` required a median training time of 205 minutes, while running the same model with FHE resulted in a median training time of 240 minutes—an increase of 17.07%. Similarly, the `cnn-qnn` models exhibited a 11.41% increase in training time, and the

`cnn-qcnn` models required 3.82% more training time. It stands out that the `cnn-qcnn` and `resnet18-qcnn` models have significantly longer training durations. This is because these models require 8 qubits instead of 4, and quantum computations were simulated on CPU. Runtimes are expected to be differ when executed on an actual quantum computer. The `FHE-resnet18-qcnn` model required a median training time of 462.30 minutes, a similar value to `FHE-resnet18-qnn`, even though it used only one client.

Interestingly, the median client round time shows only moderate increases. In some cases, it even decreases slightly when applying FHE. For example, the `Standard-resnet18` model has a median client round time of 17.82 minutes, whereas the corresponding FHE model requires only 17.50 minutes. In contrast, the central party round time consistently increases with FHE. This suggests that most of the increase in training time is due to central party-specific tasks such as parameter aggregation. This assumption is further supported by the observed encryption, decryption, and parameter aggregation times that are displayed in Table 3. Encryption and decryption of all parameters is very fast, taking between one and two seconds. However, performing calculations on homomorphically encrypted data is expensive, causing the parameter aggregation time to increase dramatically. Models without FHE typically require less than one second to aggregate parameters, as the central party simply calculates a weighted average of the 20 input matrices received from the clients. When using FHE, however, models that leverage 20 clients report a median aggregation time between 56.18 and 63.58 seconds.

Table 2: Median values for total training time, client round time, and central party round time (in minutes) by model.

| Model | Total Time | Client Round Time | Central Party Round Time |
|---|---|---|---|
| Standard-cnn | 205.06 | 9.74 | 9.99 |
| FHE-cnn | 239.86 | 9.93 | 10.63 |
| Standard-cnn-qnn | 292.01 | 13.85 | 14.23 |
| FHE-cnn-qnn | 325.33 | 14.12 | 14.82 |
| Standard-cnn-qcnn | 682.40 | 32.50 | 33.53 |
| FHE-cnn-qcnn | 708.45 | 32.75 | 33.72 |
| Standard-resnet18 | 375.02 | 17.82 | 17.99 |
| FHE-resnet18 | 401.26 | 17.50 | 18.65 |
| Standard-resnet18-qnn | 446.67 | 21.49 | 21.85 |
| FHE-resnet18-qnn | 484.15 | 21.95 | 22.68 |
| Standard-resnet18-qcnn | 848.01 | 40.91 | 41.63 |
| FHE-resnet18-qcnn* | 462.30 | 22.26 | 22.62 |

\* The `FHE-resnet18-qcnn` model was trained with only one client.

## 5.2 Increase of RAM and CPU Usage

As indicated before, the application of FHE is very memory intensive. Models without FHE required roughly 1 GiB of real RAM on the client side based on median values, as shown in Table 4. All models except `FHE-resnet18-qcnn` have approximately 2,000 parameters. Applying FHE increases the median RAM usage of clients to ~4 GiB. For example, for `resnet18`, employing FHE increased the median RAM usage from 968.88 MiB to 4,390.12 MiB, representing a 353% increase. `FHE-resnet18-qcnn`, which has 4,145 parameters, reports an even higher median RAM usage of 7,555.04 MiB. Interestingly, the presence of quantum layers does not appear to significantly affect real memory usage of clients.

On the central party side, applying FHE has an even stronger effect on real memory usage (see Table 4). While the central party requires less than 1 GiB when training a model without FHE, this value skyrockets to over 50 GiB when FHE is employed. For instance, `Standard-resnet18-qnn` requires 914.64 MiB based on median values, whereas `FHE-resnet18-qnn` requires 51.35 GiB — an increase of 5,648.44%. Even `FHE-resnet18-qcnn` reported a substantial increase of 1,355.51% (from 915.58 MiB to 13,326.36 MiB), despite `Standard-resnet18-qcnn` training with 19 more clients.

Table 3: Median values for encryption time, decryption time, and parameter aggregation time (in seconds) by model.

| Model | Encryption Time | Decryption Time | Parameter Aggregation Time |
|---|---|---|---|
| Standard-cnn | - | - | 0.03 |
| FHE-cnn | 1.68 | 1.54 | 58.45 |
| Standard-cnn-qnn | - | - | 0.03 |
| FHE-cnn-qnn | 1.53 | 1.55 | 60.06 |
| Standard-cnn-qcnn | - | - | 0.03 |
| FHE-cnn-qcnn | 1.52 | 1.64 | 63.58 |
| Standard-resnet18 | - | - | 0.01 |
| FHE-resnet18 | 1.34 | 1.51 | 56.18 |
| Standard-resnet18-qnn | - | - | 0.016 |
| FHE-resnet18-qnn | 1.32 | 1.56 | 57.85 |
| Standard-resnet18-qcnn | - | - | 0.016 |
| FHE-resnet18-qcnn* | 1.29 | 2.48 | 7.47 |

\* The `FHE-resnet18-qcnn` model was trained with only one client.

When considering RAM-related metrics, it is important to note that they were recorded throughout the entire training duration, and computational load is asynchronous in federated learning. However, even while clients train their local models, the central party consistently reported elevated RAM usage close to the displayed median values. Detailed statistics can be found in Appendix B.2.

Table 4: Median values for real RAM usage (in MiB) by model for clients and central party.

| Model | Client Real RAM | Central Party Real RAM |
|---|---|---|
| Standard-cnn | 930.04 | 816.50 |
| FHE-cnn | 3,638.55 | 51,454.87 |
| Standard-cnn-qnn | 962.34 | 821.91 |
| FHE-cnn-qnn | 3,704.29 | 52,613.89 |
| Standard-cnn-qcnn | 1,041.23 | 824.47 |
| FHE-cnn-qcnn | 3,857.98 | 54,598.11 |
| Standard-resnet18 | 968.88 | 911.90 |
| FHE-resnet18 | 4,390.12 | 51,495.80 |
| Standard-resnet18-qnn | 971.16 | 914.64 |
| FHE-resnet18-qnn | 4,375.12 | 52,577.55 |
| Standard-resnet18-qcnn | 984.95 | 915.58 |
| FHE-resnet18-qcnn* | 7,555.04 | 13,326.36 |

\* The `FHE-resnet18-qcnn` model was trained with only one client.

Regarding client CPU usage, FHE does not appear to introduce a significant impact. Each client typically used around two full CPU cores. However, models employing FHE often exhibit higher maximum CPU usage values compared to their non-FHE counterparts. In contrast, the CPU usage of central party processes is noticeably higher when models incorporate FHE for parameter encryption. Since the central party spends most of its time waiting for clients, its overall CPU usage is generally low. Nevertheless, during parameter aggregation phases, the central party experiences pronounced CPU peaks, which contribute to longer training durations (cf. Appendix B.3). Figure 2 illustrates this behavior for two exemplary runs of the `cnn` model.

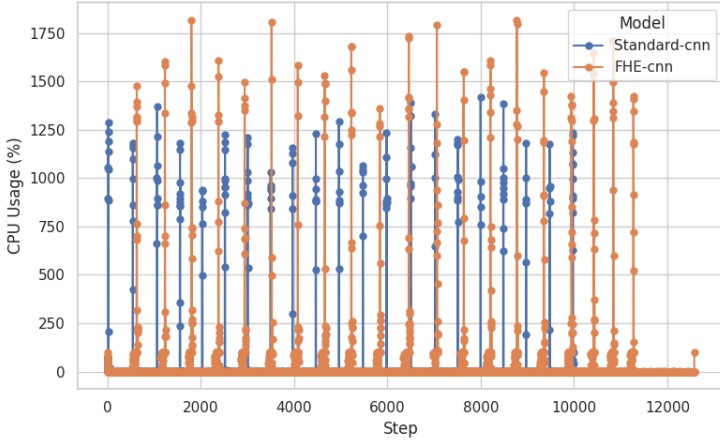

Figure 2: central party CPU usage over two exemplary runs: `FHE-cnn` and `Standard-cnn`. Each step on the x-axis represents a data point where the metric was recorded during the training process.

Table 5: Median values for communication overhead (in MiB) by model during one training round. Please note that MiB Sent refers to the parameter size sent to each individual client, whereas MiB Received depends on the total number of clients.

| Model | MiB Sent (per Client) | MiB Received (Total) |
|---|---|---|
| Standard-cnn | 0.01 | 0.18 |
| FHE-cnn | 258.76 | 13,185.30 |
| Standard-cnn-qnn | 0.01 | 0.19 |
| FHE-cnn-qnn | 264.27 | 13,465.86 |
| Standard-cnn-qcnn | 0.01 | 0.20 |
| FHE-cnn-qcnn | 280.41 | 14,288.30 |
| Standard-resnet18 | 0.01 | 0.16 |
| FHE-resnet18 | 256.76 | 13,083.33 |
| Standard-resnet18-qnn | 0.01 | 0.17 |
| FHE-resnet18-qnn | 262.26 | 13,363.84 |
| Standard-resnet18-qcnn | 0.02 | 0.33 |
| FHE-resnet18-qcnn* | 518.65 | 1,321.42 |

* The `FHE-resnet18-qcnn` model was trained with only one client.

### 5.3 Surging Communication Overhead

Our experiments demonstrate excessive communication overhead when FHE is applied, as shown in Table 5. For example, the parameter size of `Standard-cnn` models at the beginning of each training round is approximately 9 KiB. When encrypted, this size skyrockets to 258.76 MiB, representing an increase of 2,875,680%. With 20 clients, the central party receives around 13 GiB of data each round instead of just 0.18 MiB. Models with a similar number of parameters show comparable results; this includes `cnn-qnn`, `cnn-qcnn`, `resnet18`, and `resnet18-qnn`.

As previously mentioned, `FHE-resnet18-qcnn` was trained with 4,145 parameters and a single client. Consequently, the bytes sent each round doubles, while the received package size is lower in total but higher relative to the number of clients.

These results demonstrate that the communication overhead is enormous, complicating the application of FHE in QFL setups in real-world scenarios. This is especially challenging in cross-device setups, which often involve hundreds of clients (Kairouz et al., 2021).

### 5.4 Drop in Classification Performance

Considering classification metrics shown in Table 6, it becomes clear that drastically reducing the number of parameters diminishes accuracy. This is reasonable, since this work leveraged models with only 2,068 to 4,145 parameters. This illustrates the trade-off between privacy and model complexity introduced by FHE. If higher privacy standards necessitate the use of FHE, one might also reduce model complexity, which negatively impacts accuracy.

What stands out are the even lower values for central accuracy. The `Standard-cnn` reports the highest value in this category, at only 65.55%. We assume that these values would be higher if fewer clients were used. However, this metric serves as a good indicator of overall model performance since it tests the model using a separate hold-out dataset at the central party level. It might also indicate a performance drop compared to centralized learning. Applying FHE itself, however, only slightly impacted classification metrics.

Using a ResNet-18 model without quantum layers yielded promising results. The `Standard-resnet18` model achieved an aggregated F1-score of 0.89. When FHE was applied, the F1-score remained stable, and the aggregated accuracy even improved slightly. However, the central accuracy stayed low at 52.63%. This represents a partial success, demonstrating that a model leveraging FHE can be trained while maintaining moderate overhead and high classification metrics. Combining the ResNet-18 model with quantum layers did not fulfill this promise, as those models struggled to converge and hence reported low classification metrics. It is important to note that results would likely differ if quantum computations were not simulated on a CPU.

Table 6: Median values of central accuracy (in %), aggregated accuracy (in %), and aggregated F1-score by model.

| Model | Central Accuracy | Aggregated Accuracy | Aggregated F1 |
|---|---|---|---|
| Standard-cnn | 65.55 | 72.54 | 0.71 |
| FHE-cnn | – | 71.65 | 0.71 |
| Standard-cnn-qnn | 61.05 | 68.63 | 0.67 |
| FHE-cnn-qnn | – | 70.78 | 0.69 |
| Standard-cnn-qcnn | 61.43 | 68.26 | 0.66 |
| FHE-cnn-qcnn | – | 70.77 | 0.69 |
| Standard-resnet18 | 52.63 | 90.20 | 0.89 |
| FHE-resnet18 | – | 90.55 | 0.89 |
| Standard-resnet18-qnn | 44.21 | 46.00 | 0.44 |
| FHE-resnet18-qnn | – | 76.81 | 0.74 |
| Standard-resnet18-qcnn | 39.48 | 66.65 | 0.62 |
| FHE-resnet18-qcnn* | – | 76.86 | 0.75 |

* The `FHE-resnet18-qcnn` model was trained with only one client.

## 6 Discussion

This work implemented a QFL setup that enables parameter encryption using CKKS. Specifically, six different models were tested both with and without FHE. The primary goal of this work was to provide an in-depth overview of the communication and computational overhead introduced by FHE, as well as its impact on classification performance. Our findings provide guidance on the practical applicability of FHE in real-world settings and clarify the associated computational and resource demands. The objective was to assess whether FHE for parameter encryption remains a theoretical framework or can be effectively deployed in production environments. Additionally, an implementation of a QCNN within an FL setup supporting FHE was provided.

### 6.1 Findings

The overhead introduced by FHE is considerable across multiple metrics. First, median training time typically increased by 20 to 30 minutes when parameter encryption was employed. A closer analysis reveals that this increase is primarily due to longer parameter aggregation times on the central party side. However, the use of quantum layers had an even more pronounced impact on training duration. This is likely because quantum computations were simulated, so this conclusion should be interpreted with caution.

Measuring RAM and CPU usage of the client and central party processes showed that employing FHE results in a massive memory overhead and higher CPU usage peaks. Memory usage on the client side increased sharply, from roughly 1 GiB per client to about 4 GiB, when encrypting around 2,000 parameters using FHE. This makes FHE impractical for cross-device setups, as client devices may lack sufficient resources, especially as model complexity grows. As described in Section 4, we considered only models with a relatively low parameter count. Originally, we planned to deploy more complex models. However, even small fully connected layers can easily have millions of trainable parameters. For example, a self-trained CNN with 12,850,788 trainable parameters caused an out-of-memory error on the client side, despite allocating 200 GB of RAM to the process. This highlights the challenges of deploying QFL setups with FHE, as performing our experiments required a significant reduction in model complexity, despite being executed on a high-performance computing cluster. On the central party side, the effect is even more pronounced. Without FHE, the central party typically required less than 1 GiB of RAM when 20 clients participated in the learning process. With FHE, this requirement surged to over 50 GiB.

A similar effect is observed when considering communication overhead. The size of parameters for all models without FHE was 0.02 MiB or less. This means the central party sent roughly 0.02 MiB to each client and received the same amount multiplied by the number of clients. With FHE, however, this value jumped dramatically to between 256.76 MiB and 280.41 MiB for models with approximately 2,000 parameters. The ResNet-18 model combined with a QCNN, which has about twice as many parameters, reported a parameter size of 518.65 MiB. Clearly, this implies that the central party must handle a substantial amount of incoming data, especially as the number of clients increases. For models with around 2,000 parameters, the central party received more than 13 GiB of data every round.

In conclusion, although FHE enables secure parameter aggregation on the central party side, the computational and communication overhead remains too high for most real-world applications, particularly in cross-device setups. This work assumed a cross-silo environment with hospitals as clients. However, even when all parties possess substantial computational resources, the problem intensifies as more complex models need to be trained. Applying FHE uniformly to all parameters without any form of layer selection results in an inherent trade-off between privacy and model complexity.

### 6.2 Limitations

This work employed TenSEAL to enable CKKS for parameter encryption. The polynomial modulus degree and coefficient modulus bit sizes were not tuned and were instead set to the library's default values (TenSEAL, 2025). Overhead could potentially be reduced while maintaining an adequate security level if these parameters were optimally selected. One possible improvement would be to implement a function that calculates the optimal values, similar to the approach used in the FedSHE framework (Pan et al., 2024). FedSHE also proposes an encryption method that enables more efficient encryption of machine learning models by addressing the encryption length constraints of CKKS. This work simply encrypted every layer separately using untuned parameters, which increases the number of ciphertexts and consequently the communication overhead. However, it should be noted that FedSHE does not incorporate quantum layers in its implementation.

Furthermore, although computational and communication overhead were measured precisely in this work, it is important to note that clients and central partys were assigned shared resources. CPU and RAM usage were tracked by monitoring the corresponding processes, so the results remain valid. However, setting up experiments with dedicated resources for each participant would yield even more accurate measurements. Communication overhead was estimated solely by measuring the size of the trainable parameters.

It is important to note that all quantum models in this study were simulated on classical hardware. Consequently, aspects such as gradient estimation, state collapse upon measurement, and constraints on access to real quantum devices introduce substantial overhead that is not captured in our simulations. Therefore, the reported runtimes and memory usage should not be interpreted as indicative of performance on actual quantum hardware. The observations in this work primarily support conclusions regarding the overhead introduced by FHE in federated learning; they do not allow for meaningful statements about the computational cost or efficiency of federated learning systems leveraging real quantum machine learning models.

What stands out are the lower values for classification metrics resulting from reduced model complexity. Especially, the combination of ResNet-18 and quantum layers performs poorly. An in-depth analysis revealed that these models fail to converge. Zhou et al. (2023) observed a similar issue with ResNet-50 in an FL environment on the same dataset, achieving a maximum test score of only 65.32%. The authors attributed this to ResNet-50 struggling with the data heterogeneity of the dataset, which prevented convergence. Their study suggests that EfficientNet (Tan & Le, 2019) algorithms may be better suited for the experiments conducted in this work compared to ResNet-18.

To sum up, this work demonstrated that even complex QCNN models can be integrated into a QFL setup supporting parameter encryption. However, reducing the number of parameters to reduce FHE overhead has a highly negative impact on classification performance. More sophisticated methods are needed to balance security and model complexity, for example by encrypting only a subset of layers.

## 7  Conclusion and Outlook

This work provides an extensive overview of the overhead and implications introduced by FHE when integrated into a QFL framework. It offers valuable insights into the resources required to apply FHE for securing parameters, as well as the trade-offs that must be considered. Furthermore, this work is the first to present an implementation of a QCNN trained in a federated manner with parameters encrypted using CKKS.

In a scenario involving distributed training on medical data, multiple models were trained to classify brain MRI scans, encompassing a variety of classical and quantum machine learning models. Computational and communication overheads were assessed both with and without FHE by measuring CPU and RAM usage, various time-related metrics, and the size of data packets sent and received by the central party. Classification performance was evaluated using accuracy, F1-score, recall, and precision.

The experiments showed that leveraging FHE in QFL setups introduces excessive memory and communication overhead, making it potentially impractical for many real-world scenarios. This especially applies to cross-device setups. Clients typically required around four times as much RAM, while the central central party demanded up to 50 times more compared to environments without FHE. A similar pattern emerged for communication overhead: the size of roughly 2,000 parameters increased drastically from about 9 KiB to over 250 MiB. To mitigate this issue, it is desirable to keep the number of trainable parameters low, although this negatively impacts classification performance. Hence, this work demonstrates a clear trade-off between data privacy and model complexity when applying FHE. In the experiments, simple models were used to enable encryption of all layers with tolerable overhead, but these models performed poorly.

Intuitively, employing transfer learning by integrating pre-trained models like ResNet-18 with quantum layers offers a way to leverage complex models while keeping the number of trainable parameters low. This approach showed promising results, as a standalone ResNet-18 without any quantum layers achieved an F1-score of 0.89 while all parameters were encrypted. Nevertheless, adding quantum layers led to low accuracy, potentially due to data heterogeneity.

This underlines the need for more sophisticated methods to apply FHE effectively in QFL setups. There is no simple solution, such as transfer learning alone, that can fully resolve the trade-off between privacy and model complexity. Thus, for future research, it is advisable to bridge the gap between FHE in traditional FL setups and FHE in QFL environments. For example, it may not be necessary to encrypt all layers to ensure security. Instead, an algorithm could be developed to encrypt only sensitive layers, similar to the approach used in FedML-HE (Jin et al., 2024). Another promising direction is the application of clustering

methods, such as *Many Encrypted Cluster Matrices* (MECMs) (Hijazi et al., 2024), to QFL. This approach involves grouping clients into clusters so that the central party receives only one encrypted matrix per cluster. As this work demonstrated, the majority of additional training time is due to parameter aggregation; using clustering techniques could potentially reduce central party load and communication overhead. Other promising methods include optimizing CKKS parameters (as explored in Pan et al. (2024)) or combining differential privacy with FHE (as in Wu et al. (2025)). Furthermore, the limitations of this work could be addressed in future work. For instance, parameters were encrypted without optimizing CKKS settings. Exploring different pre-trained models also appears promising in order to achieve better results. Following the findings of Zhou et al. (2023), ResNet models may not be well suited for the employed dataset; instead, EfficientNet (Tan & Le, 2019) might prove more favorable. Lastly, central party and client resources were shared in this study. Though it is not expected to change the conclusions, measurements would be more precise if each participant were allocated clearly separated CPU and RAM resources.

To summarize, this work demonstrated that applying FHE in QFL setups still incurs excessive computational and communication overhead. To keep this overhead manageable, it is necessary to reduce the number of parameters, which negatively impacts classification performance - highlighting a trade-off between privacy and model complexity. Using pre-trained models like ResNet-18 showed partially promising results; however, more sophisticated techniques must be developed to integrate FHE effectively into QFL environments. This will require bridging the gap between FL and QFL research, as promising approaches already exist in FL, such as selectively encrypting sensitive layers.

## Code Availability

Our source code is available at the following URL: `https://anonymous.4open.science/r/encrypted-qfl-3461`.

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

# A   Experimental Setup

Within the appendix, a detailed overview of the experimental configuration and the collected metrics is provided.

## A.1   Collected Metrics

To evaluate the computational and communication overhead introduced by FHE and quantum layers, a comprehensive set of metrics is collected throughout the training process. These metrics capture both static properties — such as the number of trainable parameters — and dynamic measurements recorded on either the server side (e.g., parameter aggregation time) or the client side (e.g., parameter encryption time). Table 7 provides a detailed overview of all tracked metrics. The provided codebase (see 7) also provides detailed documentation of the underlying metrics.

## A.2   Experiment Configuration

This section describes the configuration used to conduct the experiments, as implemented in the corresponding code repository (see Table 8).

Table 8: Experimental configuration used for all reported results.

| Configuration Parameter | Value |
|---|---|
| Random seeds | {0, 7, 42, 420, 2025} |
| Number of experimental runs | 5 (one per random seed) |
| Training epochs per round | 10 |
| Batch size | 32 |
| Training rounds | 20 |
| Number of clients | 20* |
| Client participation per round | 100% of clients |
| Evaluation participation per round | 50% of clients |
| Data split per client | 90% training / 10% testing |
| Learning rate | 0.003 |
| Computation device | CPU (no GPU acceleration) |
| Data loading workers per client | 1 |
| Encrypted model layers | All trainable layers (when encryption enabled) |
| Quantum simulation (QNN models) | 4 qubits, 6 entangling layers |
| Quantum simulation (QCNN models) | 8 qubits, QCNN-specific architecture |
| CKKS Poly Modulus Degree | 8192 |
| CKKS Coefficient Modulus Bit Sizes | {60, 40, 40, 60} |
| CKKS Global Scale | $2^{40}$ |

\* The `FHE-resnet18-qcnn` model was trained with only one client.

## A.3   Model Architectures

This subsection describes the model architectures used in the experiments (see Tables 9 10 11 12 13 14).

Table 7: Metrics used in the analysis.

| Metric | Unit | Description |
| --- | --- | --- |
| Trainable Parameters | Integer | Number of parameters requiring gradient computation. |
| Total Training Time | Seconds | Time from server start to completion of training. |
| Server Round Time | Seconds | Duration of one server round, from sending parameters to receiving all client updates. |
| Client Round Time | Seconds | Time for a client to complete one training round. |
| Round | Integer | Number of completed training rounds. |
| Parameter Aggregation Time | Seconds | Time for the server to aggregate client parameters. |
| Encryption Time | Seconds | Time for a client to encrypt parameters. |
| Decryption Time | Seconds | Time for a client to decrypt parameters. |
| Central Accuracy | Percentage | Validation accuracy computed by the server (only for unencrypted parameters). |
| Central Loss | Float | Validation loss computed by the server (only for unencrypted parameters). |
| Aggregated Accuracy | Percentage | Client test accuracy averaged by the server. |
| Aggregated Recall | Float | Client test recall averaged by the server. |
| Aggregated Precision | Float | Client test precision averaged by the server. |
| Aggregated F1 | Float | Client test F1-score averaged by the server. |
| Aggregated Loss | Float | Client test loss averaged by the server. |
| Server Virtual Memory Usage | Megabyte | Virtual memory usage of the server process. |
| Server Real Memory Usage | Megabyte | Resident memory usage of the server process. |
| Server CPU Usage | Percentage | Average server CPU usage as a percentage of one core. |
| Client Virtual Memory Usage | Megabyte | Virtual memory usage of a client process. |
| Client Real Memory Usage | Megabyte | Resident memory usage of a client process. |
| Client CPU Usage | Percentage | Average client CPU usage as a percentage of one core. |
| Total Bytes Received | Integer | Total bytes received by the server during training, based on gradient size. |
| Total Bytes Sent | Integer | Total bytes sent by the server to one client during training, based on gradient size. |
| Bytes Sent (Round) | Integer | Bytes sent by the server to one client in a single training round, based on gradient size.. |
| Bytes Received (Round) | Integer | Bytes received by the server from all clients in a single training round, based on gradient size.. |

Table 9: Architecture of the convolutional neural network (`cnn`) used in the experiments.

| Layer | Operation | Output Shape |
|---|---|---|
| Input | RGB image | $3 \times 224 \times 224$ |
| Conv Block 1 | Conv2D + ReLU + MaxPool | $8 \times 112 \times 112$ |
| Conv Block 2 | Conv2D + ReLU + MaxPool | $16 \times 56 \times 56$ |
| Global Pooling | AdaptiveAvgPool2D | $16 \times 1 \times 1$ |
| Flatten | Reshape | $1 \times 16$ |
| Fully Connected 1 | Linear + ReLU | $1 \times 32$ |
| Fully Connected 2 | Linear | $1 \times 4$ |
| Output | Class logits | 4 |

Table 10: Architecture of the convolutional neural network with quantum layers (`cnn-qnn`) used in the experiments.

| Layer | Operation | Output Shape |
|---|---|---|
| Input | RGB image | $3 \times 224 \times 224$ |
| Conv Block 1 | Conv2D + ReLU + MaxPool | $8 \times 112 \times 112$ |
| Conv Block 2 | Conv2D + ReLU + MaxPool | $16 \times 56 \times 56$ |
| Global Pooling | AdaptiveAvgPool2D | $16 \times 1 \times 1$ |
| Flatten | Reshape | $1 \times 16$ |
| Fully Connected 1 | Linear + ReLU | $1 \times 32$ |
| Fully Connected 2 | Linear | $1 \times 4$ |
| Quantum Layer | Simple quantum circuit | 4 qubits |
| Fully Connected 3 | Linear | $1 \times 4$ |
| Output | Class logits | 4 |

Table 11: Architecture of the convolutional neural network with quantum convolutional layers (`cnn-qcnn`) used in the experiments.

| Layer | Operation | Output Shape / Qubits |
|---|---|---|
| Input | RGB image | $3 \times 224 \times 224$ |
| Conv Block 1 | Conv2D + ReLU + MaxPool | $8 \times 112 \times 112$ |
| Conv Block 2 | Conv2D + ReLU + MaxPool | $16 \times 56 \times 56$ |
| Global Pooling | AdaptiveAvgPool2D | $16 \times 1 \times 1$ |
| Flatten | Reshape | $1 \times 16$ |
| Fully Connected 1 | Linear + ReLU | $1 \times 32$ |
| Fully Connected 2 | Linear | $1 \times 8$ |
| Quantum Convolutional Layer | QCNN (parameterized quantum circuit) | 8 qubits |
| Fully Connected 3 | Linear | $1 \times 4$ |
| Output | Class logits | 4 |

Table 12: Architecture of the ResNet-18 model (`resnet18`) used in the experiments.

| Layer | Operation | Output Shape |
|---|---|---|
| Input | RGB image | $3 \times 224 \times 224$ |
| Feature Extractor | ResNet-18 (predefined) | $1 \times 512$ |
| Fully Connected | Linear | $1 \times 4$ |
| Output | Class logits | 4 |

Table 13: Architecture of the ResNet-18 model with quantum layers (`resnet18-qnn`) used in the experiments.

| Layer | Operation | Output Shape / Qubits |
|-------|-----------|----------------------|
| Input | RGB image | $3 \times 224 \times 224$ |
| Feature Extractor | ResNet-18 (predefined) | $1 \times 512$ |
| Fully Connected 1 | Linear | $1 \times 4$ |
| Quantum Layer | Parameterized quantum circuit | 4 qubits |
| Fully Connected 2 | Linear | $1 \times 4$ |
| Output | Class logits | 4 |

Table 14: Architecture of the ResNet-18 model with quantum convolutional layers (`resnet18-qcnn`) used in the experiments.

| Layer | Operation | Output Shape / Qubits |
|-------|-----------|----------------------|
| Input | RGB image | $3 \times 224 \times 224$ |
| Feature Extractor | ResNet-18 (predefined) | $1 \times 512$ |
| Fully Connected 1 | Linear | $1 \times 8$ |
| Quantum Convolutional Layer | QCNN (parameterized quantum circuit) | 8 qubits |
| Fully Connected 2 | Linear | $1 \times 4$ |
| Output | Class logits | 4 |

## B    Results

The following section provides descriptive statistics of the experimental results.

### B.1    Run Time Analysis

The following section presents detailed descriptive statistics of the results discussed in Section 5.1. All metrics, including total training time (cf. Table 15) and round times of both clients (cf. Table 16) and the central party (cf. Table 17), exhibit low standard deviations. This indicates that our results are stable across all five runs. Furthermore, the findings show that applying FHE consistently increases runtimes, with this increase primarily attributed to longer parameter aggregation times (cf. Tables 18, 19, and 20). The strongest increase in runtimes was due to quantum simulation.

Table 15: Descriptive statistics for total training time (minutes) by model. The time is measured from when the central party starts until the training is complete. Each experiment was run 5 times with different seeds. Training was conducted over 20 rounds with 20 clients.

| Model | Mean | Std | Min | 25% | 50% | 75% | Max |
|---|---|---|---|---|---|---|---|
| Standard-cnn | 205.89 | 5.06 | 199.76 | 204.39 | 205.06 | 206.50 | 213.73 |
| FHE-cnn | 239.04 | 6.08 | 229.76 | 238.11 | 239.86 | 240.89 | 246.59 |
| Standard-cnn-qnn | 289.97 | 4.53 | 283.77 | 286.73 | 292.01 | 292.78 | 294.56 |
| FHE-cnn-qnn | 324.06 | 3.76 | 317.98 | 323.24 | 325.33 | 326.01 | 327.75 |
| Standard-cnn-qcnn | 676.12 | 19.44 | 642.46 | 678.54 | 682.40 | 685.34 | 691.87 |
| FHE-cnn-qcnn | 705.11 | 10.24 | 691.14 | 699.18 | 708.45 | 708.91 | 717.89 |
| Standard-resnet18 | 375.62 | 5.71 | 369.40 | 370.82 | 375.02 | 380.76 | 382.09 |
| FHE-resnet18 | 398.80 | 15.31 | 383.30 | 383.80 | 401.26 | 406.82 | 418.84 |
| Standard-resnet18-qnn | 449.27 | 7.72 | 440.23 | 444.97 | 446.67 | 455.71 | 458.75 |
| FHE-resnet18-qnn | 483.27 | 10.31 | 469.01 | 478.57 | 484.15 | 488.11 | 496.52 |
| Standard-resnet18-qcnn | 845.53 | 7.56 | 833.55 | 843.98 | 848.01 | 848.26 | 853.85 |
| FHE-resnet18-qcnn* | 461.65 | 1.41 | 460.04 | 460.26 | 462.30 | 462.43 | 463.21 |

\* The `FHE-resnet18-qcnn` model was trained with only one client.

Table 16: Descriptive statistics for client round times (minutes) by model. The metric measures the time for a client to perform one round of training. Each experiment was run 5 times with different seeds. Training was conducted over 20 rounds with 20 clients.

| Model | Mean | Std | Min | 25% | 50% | 75% | Max |
|---|---|---|---|---|---|---|---|
| Standard-cnn | 9.71 | 0.39 | 7.21 | 9.57 | 9.74 | 9.94 | 10.47 |
| FHE-cnn | 9.90 | 0.38 | 7.84 | 9.66 | 9.93 | 10.20 | 10.53 |
| Standard-cnn-qnn | 13.69 | 0.73 | 9.11 | 13.53 | 13.85 | 14.13 | 14.83 |
| FHE-cnn-qnn | 13.98 | 0.60 | 10.59 | 13.70 | 14.12 | 14.39 | 15.85 |
| Standard-cnn-qcnn | 31.52 | 2.97 | 11.78 | 30.82 | 32.50 | 33.36 | 34.89 |
| FHE-cnn-qcnn | 32.61 | 1.21 | 23.31 | 32.25 | 32.75 | 33.29 | 35.43 |
| Standard-resnet18 | 17.87 | 0.60 | 15.92 | 17.55 | 17.82 | 18.07 | 20.30 |
| FHE-resnet18 | 16.96 | 2.33 | 4.93 | 17.15 | 17.50 | 18.17 | 19.54 |
| Standard-resnet18-qnn | 21.50 | 0.51 | 16.50 | 21.25 | 21.49 | 21.82 | 22.67 |
| FHE-resnet18-qnn | 21.90 | 0.67 | 18.83 | 21.51 | 21.95 | 22.33 | 23.40 |
| Standard-resnet18-qcnn | 40.66 | 1.23 | 31.41 | 40.33 | 40.91 | 41.45 | 42.09 |
| FHE-resnet18-qcnn* | 22.21 | 0.20 | 21.52 | 22.13 | 22.26 | 22.33 | 22.56 |

\* The `FHE-resnet18-qcnn` model was trained with only one client.

Table 17: Descriptive statistics for central party round times (minutes) by model. The metric measures the time taken to complete one training round on the central party side, from when parameters are sent to clients until all client updates are received. Each experiment was run 5 times with different seeds. Training was conducted over 20 rounds with 20 clients.

| Model | Mean | Std | Min | 25% | 50% | 75% | Max |
|-------|------|-----|-----|-----|-----|-----|-----|
| Standard-cnn | 10.02 | 0.22 | 9.52 | 9.94 | 9.99 | 10.11 | 10.48 |
| FHE-cnn | 10.58 | 0.22 | 10.14 | 10.52 | 10.63 | 10.74 | 10.87 |
| Standard-cnn-qnn | 14.17 | 0.29 | 13.70 | 13.98 | 14.23 | 14.34 | 14.84 |
| FHE-cnn-qnn | 14.83 | 0.23 | 14.51 | 14.68 | 14.82 | 14.94 | 16.05 |
| Standard-cnn-qcnn | 33.36 | 0.94 | 30.83 | 33.42 | 33.53 | 33.81 | 34.90 |
| FHE-cnn-qcnn | 33.77 | 0.55 | 33.05 | 33.44 | 33.72 | 34.11 | 35.65 |
| Standard-resnet18 | 18.22 | 0.53 | 15.92 | 17.93 | 17.99 | 18.24 | 20.30 |
| FHE-resnet18 | 18.59 | 0.65 | 17.81 | 17.87 | 18.65 | 19.13 | 19.74 |
| Standard-resnet18-qnn | 21.87 | 0.37 | 21.17 | 21.62 | 21.85 | 22.18 | 22.68 |
| FHE-resnet18-qnn | 22.73 | 0.44 | 22.00 | 22.46 | 22.68 | 22.92 | 23.67 |
| Standard-resnet18-qcnn | 41.54 | 0.39 | 40.47 | 41.25 | 41.63 | 41.92 | 42.09 |
| FHE-resnet18-qcnn* | 22.57 | 0.20 | 21.86 | 22.49 | 22.62 | 22.69 | 22.93 |

\* The `FHE-resnet18-qcnn` model was trained with only one client.

Table 18: Descriptive statistics for encryption time (seconds) by model. The metric represents the time taken by a client to encrypt all parameters. Each experiment was run 5 times with different seeds. Training was conducted over 20 rounds with 20 clients.

| Model | Mean | Std | Min | 25% | 50% | 75% | Max |
|-------|------|-----|-----|-----|-----|-----|-----|
| FHE-fednn | 1.89 | 0.93 | 0.75 | 1.22 | 1.68 | 2.28 | 7.31 |
| FHE-fedqnn | 1.82 | 0.99 | 0.75 | 1.15 | 1.53 | 2.14 | 8.74 |
| FHE-qcnn | 1.89 | 1.29 | 0.77 | 1.15 | 1.52 | 2.07 | 11.74 |
| FHE-resnet18 | 1.55 | 0.82 | 0.68 | 1.06 | 1.34 | 1.71 | 7.00 |
| FHE-resnet18-qnn | 1.58 | 0.95 | 0.68 | 1.05 | 1.32 | 1.74 | 9.53 |
| FHE-resnet18-qcnn | 1.36 | 0.29 | 1.20 | 1.26 | 1.29 | 1.33 | 2.71 |

\* The `FHE-resnet18-qcnn` model was trained with only one client.

Table 19: Descriptive statistics for decryption time (seconds) by model. The metric represents the time taken by a client to decrypt all parameters. Each experiment was run 5 times with different seeds. Training was conducted over 20 rounds with 20 clients.

| Model | Mean | Std | Min | 25% | 50% | 75% | Max |
|-------|------|-----|-----|-----|-----|-----|-----|
| FHE-fednn | 1.47 | 0.35 | 0.00 | 1.44 | 1.54 | 1.61 | 2.28 |
| FHE-fedqnn | 1.49 | 0.36 | 0.00 | 1.47 | 1.55 | 1.63 | 2.48 |
| FHE-qcnn | 1.57 | 0.37 | 0.00 | 1.54 | 1.64 | 1.72 | 2.29 |
| FHE-resnet18 | 1.45 | 0.45 | 0.00 | 1.41 | 1.51 | 1.58 | 13.23 |
| FHE-resnet18-qnn | 1.49 | 0.35 | 0.00 | 1.46 | 1.56 | 1.63 | 2.17 |
| FHE-resnet18-qcnn | 2.38 | 0.54 | 0.00 | 2.47 | 2.48 | 2.48 | 2.61 |

\* The `FHE-resnet18-qcnn` model was trained with only one client.

Table 20: Descriptive statistics for parameter aggregation time (seconds) by model. The metric represents the time taken for the central party to aggregate received parameters. Each experiment was run 5 times with different seeds. Training was conducted over 20 rounds with 20 clients.

| Model | Mean | Std | Min | 25% | 50% | 75% | Max |
|---|---|---|---|---|---|---|---|
| Standard-fednn | 0.0253 | 0.0005 | 0.0246 | 0.0250 | 0.0252 | 0.0254 | 0.0286 |
| FHE-fednn | 60.4943 | 6.1601 | 55.2537 | 57.3378 | 58.4464 | 61.8221 | 102.7924 |
| Standard-fedqnn | 0.0343 | 0.0010 | 0.0332 | 0.0338 | 0.0341 | 0.0346 | 0.0411 |
| FHE-fedqnn | 60.6507 | 2.9178 | 56.8111 | 58.5955 | 60.0616 | 62.2279 | 77.8310 |
| Standard-qcnn | 0.0330 | 0.0023 | 0.0271 | 0.0320 | 0.0337 | 0.0342 | 0.0405 |
| FHE-qcnn | 64.0572 | 2.1926 | 60.0814 | 63.0356 | 63.5784 | 65.0797 | 71.7844 |
| Standard-resnet18 | 0.0066 | 0.0001 | 0.0064 | 0.0065 | 0.0066 | 0.0067 | 0.0071 |
| FHE-resnet18 | 55.9034 | 5.6232 | 44.5365 | 55.0848 | 56.1819 | 57.2370 | 70.9698 |
| Standard-resnet18-qnn | 0.0158 | 0.0003 | 0.0154 | 0.0156 | 0.0157 | 0.0159 | 0.0182 |
| FHE-resnet18-qnn | 59.6153 | 5.1504 | 54.5749 | 56.9458 | 57.8500 | 59.8854 | 87.1561 |
| Standard-resnet18-qcnn | 0.0160 | 0.0003 | 0.0157 | 0.0159 | 0.0159 | 0.0160 | 0.0168 |
| FHE-resnet18-qcnn* | 7.5641 | 0.4874 | 7.0785 | 7.4052 | 7.4735 | 7.5593 | 9.7281 |

* The `FHE-resnet18-qcnn` model was trained with only one client.

## B.2 RAM Usage

As shown in Section 5.2, FHE has a significant impact on RAM usage on both the client and central party sides. Although FL is asynchronous, FHE increases RAM consumption throughout the entire training process, as indicated by the descriptive statistics for client RAM usage (cf. Table 21) and central party RAM usage (cf. Table 22).

Table 21: Descriptive statistics for client resident (real) RAM usage (MiB) by model. Each experiment was run 5 times with different seeds. Training was conducted over 20 rounds with 20 clients.

| Model | Mean | Std | Min | 25% | 50% | 75% | Max |
|---|---|---|---|---|---|---|---|
| Standard-fednn | 941.77 | 57.39 | 345.28 | 904.00 | 930.04 | 968.83 | 1,142.82 |
| FHE-fednn | 3,628.74 | 702.12 | 89.50 | 3,566.60 | 3,638.55 | 3,737.03 | 5,612.05 |
| Standard-fedqnn | 966.43 | 68.60 | 48.91 | 919.20 | 962.34 | 1,017.84 | 1,131.27 |
| FHE-fedqnn | 3,719.00 | 672.80 | 886.36 | 3,628.05 | 3,704.29 | 3,926.04 | 5,706.62 |
| Standard-qcnn | 1,031.49 | 60.98 | 818.02 | 982.61 | 1,041.23 | 1,084.65 | 1,162.02 |
| FHE-qcnn | 3,785.47 | 663.99 | 820.05 | 3,753.08 | 3,857.98 | 3,941.46 | 5,915.72 |
| Standard-resnet18 | 971.83 | 53.83 | 261.87 | 934.96 | 968.88 | 1,001.76 | 1,196.94 |
| FHE-resnet18 | 4,313.06 | 780.83 | 901.20 | 4,307.96 | 4,390.12 | 4,460.92 | 6,385.66 |
| Standard-resnet18-qnn | 977.19 | 47.89 | 857.77 | 943.34 | 971.16 | 1,006.78 | 1,194.34 |
| FHE-resnet18-qnn | 4,285.67 | 776.32 | 936.56 | 4,282.09 | 4,375.12 | 4,482.63 | 6,465.65 |
| Standard-resnet18-qcnn | 983.56 | 46.47 | 392.87 | 951.30 | 984.95 | 1,009.42 | 1,222.64 |
| FHE-resnet18-qcnn | 7,277.89 | 1,463.03 | 994.48 | 7,512.93 | 7,555.04 | 7,601.73 | 10,813.16 |

* The `FHE-resnet18-qcnn` model was trained with only one client.

Table 22: Descriptive statistics for central party resident (real) RAM usage (MiB) by model. Each experiment was run 5 times with different seeds. Training was conducted over 20 rounds with 20 clients.

| Model | Mean | Std | Min | 25% | 50% | 75% | Max |
|-------|------|-----|-----|-----|-----|-----|-----|
| Standard-fednn | 811.41 | 24.74 | 16.00 | 799.91 | 816.50 | 820.14 | 897.27 |
| FHE-fednn | 50,284.38 | 11,859.14 | 119.00 | 51,249.21 | 51,454.87 | 53,189.10 | 66,019.55 |
| Standard-fedqnn | 817.63 | 14.75 | 780.48 | 808.06 | 821.91 | 829.73 | 897.57 |
| FHE-fedqnn | 51,260.06 | 11,280.57 | 375.68 | 52,377.88 | 52,613.89 | 53,640.09 | 66,229.84 |
| Standard-qcnn | 820.62 | 21.66 | 115.50 | 817.06 | 824.47 | 831.13 | 902.35 |
| FHE-qcnn | 52,391.44 | 12,431.80 | 879.48 | 54,400.93 | 54,598.11 | 54,756.95 | 69,577.01 |
| Standard-resnet18 | 923.29 | 39.66 | 308.15 | 903.37 | 911.90 | 923.29 | 1,103.36 |
| FHE-resnet18 | 50,496.97 | 11,554.64 | 404.07 | 50,930.93 | 51,495.80 | 52,972.50 | 66,683.84 |
| Standard-resnet18-qnn | 917.89 | 22.23 | 480.34 | 910.29 | 914.64 | 923.88 | 1,157.07 |
| FHE-resnet18-qnn | 50,780.88 | 11,554.17 | 530.07 | 52,220.18 | 52,577.55 | 52,958.82 | 67,170.53 |
| Standard-resnet18-qcnn | 916.01 | 14.25 | 881.81 | 909.89 | 915.58 | 928.73 | 1,049.42 |
| FHE-resnet18-qcnn* | 12,922.53 | 2,798.31 | 894.67 | 13,151.06 | 13,326.36 | 13,997.60 | 17,150.88 |

\* The `FHE-resnet18-qcnn` model was trained with only one client.

## B.3 CPU Usage

The descriptive statistics of Table 23 and 24 complement the results of Section 5.2. CPU related metrics are sensitive to the asynchronous nature of FL. However, applying FHE often resulted in higher CPU usage peaks. Quantum simulation is rather memory-intensive and therefore does not have a clear impact on CPU usage.

Table 23: Descriptive statistics for client CPU usage (%) by model. CPU usage is reported as a percentage of one core (for example, 200% indicates full usage of two cores). Each experiment was run 5 times with different seeds. Training was conducted over 20 rounds with 20 clients.

| Model | Mean | Std | Min | 25% | 50% | 75% | Max |
|-------|------|-----|-----|-----|-----|-----|-----|
| Standard-cnn | 236.83 | 173.27 | 2.00 | 109.90 | 189.90 | 313.70 | 1,510.50 |
| FHE-cnn | 225.44 | 159.07 | 1.90 | 105.90 | 191.80 | 311.70 | 1,565.90 |
| Standard-cnn-qnn | 233.10 | 174.19 | 2.00 | 101.90 | 191.80 | 313.70 | 1,136.80 |
| FHE-cnn-qnn | 230.53 | 183.92 | 2.00 | 99.90 | 185.80 | 319.53 | 2,086.10 |
| Standard-cnn-qcnn | 241.45 | 212.98 | 2.00 | 97.90 | 175.80 | 309.70 | 1,582.60 |
| FHE-cnn-qcnn | 239.11 | 197.73 | 2.00 | 97.90 | 179.80 | 332.70 | 1,256.70 |
| Standard-resnet18 | 236.04 | 138.52 | 2.00 | 129.90 | 209.80 | 315.70 | 987.00 |
| FHE-resnet18 | 223.59 | 184.30 | 2.00 | 103.90 | 187.70 | 289.25 | 2,678.90 |
| Standard-resnet18-qnn | 236.45 | 138.11 | 2.00 | 133.90 | 213.80 | 309.70 | 925.10 |
| FHE-resnet18-qnn | 239.59 | 169.47 | 2.00 | 117.90 | 203.80 | 317.65 | 1,831.50 |
| Standard-resnet18-qcnn | 242.83 | 197.46 | 2.00 | 99.73 | 185.80 | 323.70 | 1,292.10 |
| FHE-resnet18-qcnn | 2,004.46 | 375.03 | 2.00 | 1,880.20 | 2,032.00 | 2,301.90 | 4,111.90 |

\* The `FHE-resnet18-qcnn` model was trained with only one client.

Table 24: Descriptive statistics for central party CPU usage (%) by model. CPU usage is reported as a percentage of one core (for example, 200% indicates full usage of two cores). Each experiment was run 5 times with different seeds. Training was conducted over 20 rounds with 20 clients.

| Model | Mean | Std | Min | 25% | 50% | 75% | Max |
|---|---|---|---|---|---|---|---|
| Standard-cnn | 231.51 | 441.39 | 1.90 | 2.00 | 2.00 | 18.00 | 1,388.70 |
| FHE-cnn | 157.68 | 316.51 | 2.00 | 33.95 | 97.90 | 99.90 | 1,824.10 |
| Standard-cnn-qnn | 206.59 | 418.29 | 2.00 | 2.00 | 2.00 | 2.00 | 1,390.60 |
| FHE-cnn-qnn | 155.10 | 342.56 | 2.00 | 2.00 | 97.90 | 99.90 | 1,839.80 |
| Standard-cnn-qcnn | 283.35 | 569.37 | 2.00 | 2.00 | 2.00 | 2.00 | 1,652.50 |
| FHE-cnn-qcnn | 154.45 | 372.78 | 2.00 | 2.00 | 38.00 | 99.90 | 1,874.00 |
| Standard-resnet18 | 748.60 | 938.92 | 2.00 | 2.00 | 2.00 | 1,853.45 | 2,391.40 |
| FHE-resnet18 | 197.83 | 420.44 | 2.00 | 2.00 | 97.90 | 99.90 | 2,321.40 |
| Standard-resnet18-qnn | 614.88 | 877.82 | 2.00 | 2.00 | 2.00 | 1,744.10 | 2,133.90 |
| FHE-resnet18-qnn | 162.55 | 358.05 | 2.00 | 2.00 | 97.90 | 99.90 | 2,249.30 |
| Standard-resnet18-qcnn | 356.86 | 747.61 | 2.00 | 2.00 | 2.00 | 2.00 | 2,248.20 |
| FHE-resnet18-qcnn* | 20.66 | 68.55 | 2.00 | 2.00 | 2.00 | 2.00 | 771.10 |

\* The `FHE-resnet18-qcnn` model was trained with only one client.

## B.4 Classification Performance

Due to the low complexity of the models employed, classification performance was relatively poor. However, applying FHE had only a very slight negative effect. Leveraging ResNet-18 was proposed as a way to use more complex models while keeping the number of trainable parameters low, making it ideal for QFL setups. This approach showed promising results when no quantum layers were integrated (see the `resnet-18` models in Table 25). However, when ResNet-18 was combined with quantum layers, the models struggled to converge.

Table 25: Median values for classification metrics by model. The metrics include central accuracy measured on the central party side using a predefined hold-out dataset as well as aggregated values for accuracy, F1-score, recall, and precision. Aggregated means that each client calculates these metrics on its local test dataset, and the central party then computes a weighted average. Each experiment was run 5 times with different seeds. Training was conducted over 20 rounds with 20 clients.

| Model | Centr. Acc. | Agg. Acc. | Agg. F1 | Agg. Recall | Agg. Prec. |
|---|---|---|---|---|---|
| Standard-cnn | 65.55 | 72.54 | 0.71 | 0.72 | 0.73 |
| FHE-cnn | – | 71.65 | 0.71 | 0.72 | 0.73 |
| Standard-cnn-qnn | 61.05 | 68.63 | 0.67 | 0.69 | 0.70 |
| FHE-cnn-qnn | – | 70.78 | 0.69 | 0.70 | 0.69 |
| Standard-cnn-qcnn | 61.43 | 68.26 | 0.66 | 0.67 | 0.68 |
| FHE-cnn-qcnn | – | 70.77 | 0.69 | 0.71 | 0.71 |
| Standard-resnet18 | 52.63 | 90.20 | 0.89 | 0.90 | 0.90 |
| FHE-resnet18 | – | 90.55 | 0.89 | 0.90 | 0.90 |
| Standard-resnet18-qnn | 44.21 | 46.00 | 0.44 | 0.48 | 0.54 |
| FHE-resnet18-qnn | – | 76.81 | 0.74 | 0.76 | 0.77 |
| Standard-resnet18-qcnn | 39.48 | 66.65 | 0.62 | 0.62 | 0.63 |
| FHE-resnet18-qcnn* | – | 76.86 | 0.75 | 0.76 | 0.75 |

\* The `FHE-resnet18-qcnn` model was trained with only one client.

