# OpenReview forum: "Understanding the Resource Cost of Fully Homomorphic Encryption in Quantum Federated Learning"
_TMLR — Rejected by TMLR_

### Review · Reviewer_Dspu · 2026-01-21

**Summary Of Contributions:**

This paper makes a limited, primarily empirical contribution by implementing a quantum federated learning (QFL) pipeline in which model parameters are fully encrypted using CKKS-based fully homomorphic encryption (FHE), and by systematically measuring the resulting computational, memory, and communication overhead. Through experiments on several small-scale classical and hybrid quantum–classical models, including a QCNN, the study shows that naively applying FHE to all parameters leads to prohibitive memory usage and extreme communication costs, forcing aggressive model simplification and thereby degrading predictive performance.

**Audience:**

No

**Audience Explanation:**

The reviewer may think the findings are obvious, and the conducted experiments are under carefully controlled conditions.

**Claims And Evidence:**

Yes

**Claims Explanation:**

The work does not introduce new learning algorithms, cryptographic techniques, or theoretical insights, but rather confirms that current FHE technology remains impractical for realistic QFL deployments, highlighting an inherent trade-off between strong cryptographic privacy and model complexity. However, as discussed by the authors, the HE-based schemes used can be further optimized. Thus, the efficiency of applying HE might not be accurately assessed.

**Requested Changes:**

- It is not clear whether the emerged communication overhead is due to the quantum learning or from HE or both. The reviewer is not clear about the motivation of the studied scenario. Why focus on the quantum FL?
- It is unclear which specific aggregation rule was applied in the server. If using FedAvg, the reviewer may think that applying PHE can support the aggregation with less cost.
- A recent study shows that using a modified CKKS-based scheme with controlable algorithm and hardware acceleration can somewhat alleviate the extra cost from HE (Towards compute-efficient Byzantine-robust federated learning with fully homomorphic encryption). Although it is not a quantum learning task, the reviewer may think that such techniques can be further considered.

---

> ### Author Response · Authors · 2026-02-04
> **Authors' Response to Reviewer Dspu (I)**
>
> # Response
>
> We thank the reviewer for the feedback and for raising important questions about the motivation behind the study, the technical setup, and the relevance of the results. We appreciate the opportunity to clarify these points.
>
> ## Motivation of Our Study
> >The reviewer may think the findings are obvious, and the conducted experiments are under carefully controlled conditions.
>
> First, regarding the interest to the TMLR audience and the concern that the findings may appear "obvious": While it is generally understood that FHE introduces significant overhead, reproducible, quantitatively grounded studies that clearly document the resources required to deploy FHE in federated learning systems are scarce. This work aims to provide a transparent, reproducible, end-to-end assessment of memory and communication requirements in realistic FL and QFL pipelines. We believe that this level of detail is valuable for researchers and practitioners who are evaluating the feasibility of privacy-preserving FL systems, including those in the TMLR community.
>
> ## Communication Overhead
> >It is not clear whether the emerged communication overhead is due to the quantum learning or from HE or both. The reviewer is not clear about the motivation of the studied scenario. Why focus on the quantum FL?
>
> Regarding the source of the communication overhead, we agree that this point required clearer articulation. The overhead observed in our experiments is primarily introduced by FHE. However, since the quantum computations were fully simulated, and neither hardware execution latencies nor noise effects were considered, our results cannot provide meaningful insights into overhead originating from the quantum component. We have revised the manuscript to explicitly state that our conclusions pertain to the cost of FHE in FL and QFL settings rather than to quantum learning itself.
>
> The focus on QFL is motivated by the fact that certain quantum machine learning models can achieve competitive performance with a relatively small number of trainable parameters. Since parameter count is a dominant factor in FHE-related overhead, QML is a potentially attractive model class when privacy-preserving FL is required. Our results show that this commonly assumed benefit does not consistently materialize. We believe that transparently documented negative results provide significant value to the community by preventing redundant efforts and helping steer research toward more effective approaches.
>
> ## Aggregation Rule
> >It is unclear which specific aggregation rule was applied in the server. If using FedAvg, the reviewer may think that applying PHE can support the aggregation with less cost.
>
> Regarding the aggregation rule, we clarify that the server uses the standard FedAvg method. This information has been added to the manuscript. We selected the CKKS scheme because it is considered an efficient algorithm for privacy-preserving FL systems [2].
>
> ## Recent Work
> >A recent study shows that using a modified CKKS-based scheme with controlable algorithm and hardware acceleration can somewhat alleviate the extra cost from HE (Towards compute-efficient Byzantine-robust federated learning with fully homomorphic encryption). Although it is not a quantum learning task, the reviewer may think that such techniques can be further considered.
>
> We thank the reviewer for bringing the recent work of [3] to our attention. We agree that this study is highly relevant and have included it in our literature review. Their approach relies on hardware acceleration and advanced, GPU-based cryptographic implementations to mitigate overhead. In contrast, our work provides a general, hardware-agnostic overview of the resource requirements for FHE in FL and QFL environments. We view these approaches as complementary, especially since our work also incorporates quantum routines.
>
> In summary, we revised the manuscript to:
> 1. clearly attribute the observed overhead to FHE;
> 2. explicitly describe the aggregation mechanism;
> 3. motivate and evaluate the hypothesis that QFL’s reduced parameter count can mitigate FHE-induced overhead; and
> 4. position our contribution with respect to recent advances in efficient FHE-based FL.
> We believe these changes clarify the work and highlight its relevance for the TMLR audience.
>
> # Summary of Revisions
> - Gradient Aggregation: Clarified that FedAvg was used to aggregate model updates. See Chapter 4, p. 5.
> - Related Work Update: Included Jiang et al. (2025) in the related works section. See Chapter 2, p. 2.

---

> > ### Comment · Reviewer_Dspu · 2026-02-12
> >
> > Thanks for the effort. However, I may still think the insights from the current version are limited. As explained by the authors, I think we may all agree that the overhead is largely from HE, while QML may have a weak connection with it. To be clearer, it seems that we don't know any specific patterns from QML, or the provided results are directly applying a ckks based rule to enable aggregation, which is also not the newest one for efficiency. Also, to enable a more accurate estimation of the resource cost or to provide a benchmark, the reviewer thinks an appropriate encryption scheme should be chosen. In addition, since the authors say they use FedAVG for aggregation, which assumes there are no malicious clients in the system model, then a lightweight encryption, like PHE, should be tested.

---

> ### Author Response · Authors · 2026-02-04
> **Authors' Response to Reviewer Dspu (II)**
>
> # References
> [1] Yao Pan, Zheng Chao, Wang He, Yang Jing, Li Hongjia, and Wang Liming. Fedshe: privacy preserving and efficient federated learning with adaptive segmented ckks homomorphic encryption. Cybersecurity, 7(1): 40, 2024.
>
> [2] Lei Jiang and Lei Ju. Fhebench: Benchmarking fully homomorphic encryption schemes, 2022. URL https://arxiv.org/abs/2203.00728.
>
> [3] Jiang, S., Yang, H., Xie, Q. et al. Towards compute-efficient Byzantine-robust federated learning with fully homomorphic encryption. Nat Mach Intell 7, 1657–1668 (2025). https://doi.org/10.1038/s42256-025-01107-6

---

### Review · Reviewer_5peD · 2026-01-26

**Summary Of Contributions:**

The paper investigates the resource requirements for privacy-preserving measures in federated machine learning and quantum machine learning. Specifically, they use fully homomorphic encription by encrypting the parameters with the CKKS scheme. The setups are tested on a brain tumor dataset and it is found, that encryption for the considered model sizes around 2.000-4.000 parameters introduces enormous overhead. This partially can be mitigated by compressing models and thereby reducing parameter counts, which however also reduces performance, so there seems to be an balance to be found. Based on these findings, the authors advocate for additional research into the safety of federated quantum machine learning, as identifying more suitable techniques is required.

First of all, I want to thank the authors for submitting their work to TMLR, as safety considerations in machine learning and also quantum machine learning are certainly a timely topic.

However, from my perspective I see two fundamental problems with the way the study was constructed and the evaluation of the findings (I refer to concrete text passages in the boxes below):

1) From the perspective of realistic quantum computing, the study setup has major flaws, especially as strong emphasis is put on evaluating statistics like runtime, memory, etc.. While the authors correctly state that the quantum networks in the study are simulated on classical hardware (using PennyLane), the effects of this vs. execution on real hardware are nor sufficiently worked out. Firstly, latencies when using federated quantum systems are not at all taken into account, which likely are significant. Secondly, the execution times on quantum hardware play a major role (statements like "8 qubits can be run faster on quantum hardware than they can be simulated" do not apply), in particular, if gradients are to be computed: On quantum devices, there exists no equivalent of backpropagation, which requires to default to something like the parameter-shift rule, i.e. estimation of every individual parameter with finite differences. This introduces an enormous overhead, which very likely dwarfs any encryption overhead cited in this paper.

2) I think the paper does not really make statements specifically to (federated) quantum machine learning, but much more federated machine learning in general, where already a lot of prior work exists. For example, the models incorporating quantum components consistently under-perform the fully classical setups, see Table 6. Reducing the parameters increases this difference even further, to the point where one model with quantum components is not functional at all. This casts doubt upon the selection of quantum approaches, if these are not at least on par with the classical version -> why should one go into the enormous overhead of using quantum-enhanced models if there is not even an potential advantage? Moreover: If the overhead of encrypting the models is so enormous for about 2-4000 parameters, then this of course also applies to the classical models, not only the quantum ones. Summarized, I don't think that the results allow any statements specifically on federated QML, apart from that the uses QML models seem not to be suitable for the selected task.

**Additional Comments:**

I want to note, that my expertise regarding this paper lies in the fields of quantum computing and quantum machine learning. Consequently, I am focusing my review on this, and not directly address the technicalities of the encryption protocol.

**Audience:**

No

**Audience Explanation:**

- As argued above, I think due to the study design and the quantum models nor performing well at all on the used dataset the findings are unfortunately not relevant to the classical (federated) learning community, nor to the quantum machine learning community.

**Broader Impact Concerns:**

--

**Claims And Evidence:**

No

**Claims Explanation:**

- As argued above, I think the study setup is not suitable for evaluating properties like runtime or memory usage for QML models, as significant factors were not taken into consideration
- The results do not allow for statements particular on quantum machine learning, so I think all statements are general and do not really apply to "quantum federated learning"

**Requested Changes:**

In the following I highlight a few text passages where I see caveats (in large parts these underline issue 1 and 2 described above):

3: "These parameters are iteratively updated using conventional machine learning methods on a classical computer." (page 3): This is true if everything is simulated on a classical computer, as done in the study. However, if looking for actual quantum advantage, that one hast to at least think about running on actual hardware. This introduces the above described problems of getting access to the gradients due to state collapse upon measurement. While there exist methods to do this, they introduce significant overhead compared to training classical NNs via backpropagation.

4: "This is because these models require 8 qubits instead of 4, and quantum computations were simulated on CPU. Runtimes are expected to be faster when executed on an actual quantum computer." (page 6) The last statement does not hold under realistic assumptions. In fact, simulating either 4 or 8-qubit systems is very efficient with simulators like PennyLane. In principle, one has to evolve a state consisting of 2^4 and 2^8 elements to fully describe the quantum system. While the second one is admittedly significantly larger, due to implementation overhead the performance difference in most simulators does not show significantly until 12 to 14 qubits. The system sizes where one can expect actual runtime improvements on quantum hardware are even larger, estimated at least at 25 to 30 qubits.

5: "Combining the ResNet-18 model with quantum layers did not fulfill this promise, as those models struggled to converge and hence reported low classification metrics. It is important to note that results would likely differ if quantum computations were not simulated on a CPU." (page 10): This for me again highlights the problematic selection of the quantum routine, as it seems not suited for the considered task at all. In this formulation, I agree with the second sentence. However, the runtime would certainly swing much more into favor of the fully-classical approach.

6: "However, the use of quantum layers had an even more pronounced impact on training duration. This is likely because quantum computations were simulated, so this conclusion should be interpreted with caution." (page 10): Certainly should be interpreted with caution, but runtime-wise executing on hardware certainly wouldn't make things better, not even talking about noise at this point.


Overall, I want to emphasize that I don't argue against quantum machine learning in general here. I just think that the considered models should at least show promising behavior (on par with classical models) on the considered task to allow for meaningful statements.
Additionally,  if talking about wall-clock measures, things like execution on quantum hardware and estimating gradients must be taken into account (not necessarily by execution on actual quantum hardware, but at least extrapolation thereof)

---

> ### Author Response · Authors · 2026-02-04
> **Authors' Response to Reviewer 5peD (I)**
>
> # Response
>
> The authors thank the reviewer for the thorough review and, in particular, for carefully discussing the aspects of quantum machine learning. We appreciate the opportunity to clarify the scope and intent of our work, as well as refine the corresponding claims.
>
> ## Rationale for QML/QFL in Our Study
> >As argued above, I think due to the study design and the quantum models nor performing well at all on the used dataset the findings are unfortunately not relevant to the classical (federated) learning community, nor to the quantum machine learning community.
>
> >Overall, I want to emphasize that I don't argue against quantum machine learning in general here. I just think that the considered models should at least show promising behavior (on par with classical models) on the considered task to allow for meaningful statements. Additionally, if talking about wall-clock measures, things like execution on quantum hardware and estimating gradients must be taken into account (not necessarily by execution on actual quantum hardware, but at least extrapolation thereof)
>
> First, we agree that our study is not suitable for making strong claims about the intrinsic runtime or memory advantages of quantum machine learning, particularly with regard to potential quantum advantage. Our primary objective is not to evaluate quantum speedups, but rather to assess the practical overhead introduced by FHE in FL settings, including those that incorporate quantum machine learning components.
>
> We included QML and QFL in this study because quantum models can, in some cases, achieve competitive performance with a comparatively small number of trainable parameters. This property is particularly relevant in the context of FHE, where the parameter count dominantly drives memory and communication overhead. Thus, our results are relevant to quantum federated learning not because they demonstrate quantum advantage but because they explore whether QML architectures could make FHE-based FL deployments more feasible despite their high overhead. We now emphasize this rationale more clearly in the revised version.

---

> > ### Author Response · Authors · 2026-02-04
> > **Authors' Response to Reviewer 5peD (II)**
> >
> > ## Clarification on Quantum Simulations
> > >3: "These parameters are iteratively updated using conventional machine learning methods on a classical computer." (page 3): This is true if everything is simulated on a classical computer, as done in the study. However, if looking for actual quantum advantage, that one hast to at least think about running on actual hardware. This introduces the above described problems of getting access to the gradients due to state collapse upon measurement. While there exist methods to do this, they introduce significant overhead compared to training classical NNs via backpropagation.
> >
> > >4: "This is because these models require 8 qubits instead of 4, and quantum computations were simulated on CPU. Runtimes are expected to be faster when executed on an actual quantum computer." (page 6) The last statement does not hold under realistic assumptions. In fact, simulating either 4 or 8-qubit systems is very efficient with simulators like PennyLane. In principle, one has to evolve a state consisting of 2^4 and 2^8 elements to fully describe the quantum system. While the second one is admittedly significantly larger, due to implementation overhead the performance difference in most simulators does not show significantly until 12 to 14 qubits. The system sizes where one can expect actual runtime improvements on quantum hardware are even larger, estimated at least at 25 to 30 qubits.
> >
> > >6: "However, the use of quantum layers had an even more pronounced impact on training duration. This is likely because quantum computations were simulated, so this conclusion should be interpreted with caution." (page 10): Certainly should be interpreted with caution, but runtime-wise executing on hardware certainly wouldn't make things better, not even talking about noise at this point.
> >
> > Regarding runtime- and simulation-related comments, we agree that our experimental setup does not support conclusions about execution speed or runtime advantages of quantum models on actual hardware. We have therefore revised or removed statements suggesting improved runtimes on near-term quantum devices and clarified that our measurements are limited to simulated execution. We further acknowledge that transitioning from simulation to real quantum hardware introduces additional challenges, which are beyond the scope of this work. Accordingly, we now explicitly state that our results do not allow for meaningful conclusions about runtime or overhead contributions from the quantum execution itself.
> >
> > ## Model Performance
> > >5: "Combining the ResNet-18 model with quantum layers did not fulfill this promise, as those models struggled to converge and hence reported low classification metrics. It is important to note that results would likely differ if quantum computations were not simulated on a CPU." (page 10): This for me again highlights the problematic selection of the quantum routine, as it seems not suited for the considered task at all. In this formulation, I agree with the second sentence. However, the runtime would certainly swing much more into favor of the fully-classical approach.
> >
> > Regarding model performance, we acknowledge that the selected quantum routines performed poorly on the chosen dataset and struggled to converge in some cases. The routines were chosen based on prior work (e.g., [3]), and we intentionally report the results as obtained, together with the full source code, to ensure transparency and reproducibility and to mitigate publication bias. This highlights that incorporating quantum routines does not automatically lead to high-performing models, which we believe is a valuable and honest observation for the community.
> >
> > The main contribution of this work is a detailed and reproducible quantitative analysis of the resource requirements imposed by Fully Homomorphic Encryption in federated learning setups. Unlike prior work that treats overhead largely at a qualitative level [1][2][3][4], we provide concrete measurements of memory, communication, and training costs across different model architectures, enabling a more realistic assessment of practical feasibility. In this context, we explore quantum machine learning models, which can sometimes achieve competitive performance with comparatively fewer trainable parameters—a property that is particularly relevant for reducing FHE-induced overhead in federated learning workflows. In addition, we present, to the best of our knowledge, the first QCNN trained in a federated setting with CKKS-encrypted parameters.
> >
> > We believe the revised manuscript more clearly delineates which conclusions can and cannot be drawn and strengthens the paper's contribution to the machine learning and privacy-preserving learning communities.

---

> > > ### Comment · Reviewer_5peD · 2026-02-10
> > > **Response to: Clarification on Quantum Simulations**
> > >
> > > I appreciate the author's effort to more clearly represent in the paper that for the quantum models there are other things that also must be taken into consideration.
> > >
> > > While the formulations are now less-problematic from my point of view, I think for a study on actual runtimes etc. the actual effects can not be abstracted away, i.e. at least estimation have to be taken into account. Otherwise, the comparisons have basically no practical relevance.

---

> > > ### Comment · Reviewer_5peD · 2026-02-10
> > > **Response to: Model Performance**
> > >
> > > I appreciate the authors effort to clarify that point.
> > >
> > > From my perspective, the main problem is still the one I already described in my first reply, i.e. that the work does not use quantum models that actually show the potential of achieving on par or even improved performance with significant reduced parameter count.

---

> > ### Comment · Reviewer_5peD · 2026-02-10
> > **Response to: Rationale for QML/QFL in Our Study**
> >
> > I thank the authors for considering and adressing my concerns regarding the scope of the paper.
> >
> > I aggree and support the rational, that in the context of FHE particular quantum models might be beneficial over classical ones due to the potentially smaller parameter count (if this benefit scales to larger and relevant system sizes due to trainability issues like barren plateaus is an open debate, but that is more an underlying question about the concept of QML and is fine to be not explicitly discussed in this work).
> >
> > However, for this rational to be consistent I still think suitable quantum models must be used. In particular, as also discussed below, the quantum models do not perform particularly well on the considered setup. While I understand the argument that models were just taken from another paper, I think this lack of performance drastically weakens above line of argumentation. What rather should be done is selcting quantum models that acutally show a promising behavior on the considered task, and do so with a clearly reduced parameter count. Currently, the hybrid quantum models contain even more parameters than their purely classical counterpart, so the argument is not consistent.

---

> ### Author Response · Authors · 2026-02-04
> **Authors' Response to Reviewer 5peD (III)**
>
> # Summary of Revisions
> - VQC Parameter Updates: Clarified that parameters of our VQCs are iteratively updated using conventional machine learning methods on a classical computer, as our experiments are simulations. See Chapter 3.1, p. 3.
> - Quantum Runtime Statements: Corrected statements regarding runtimes of quantum simulations versus actual quantum computers with small numbers of qubits. See Chapter 5.1, p. 7.
> - Quantum Hardware Limitations: Added a note that our study does not capture overhead from real quantum hardware, e.g., gradient estimation or state collapse upon measurement. See Chapter 6.2, p. 12.
>
> # References
> [1] Yao Pan, Zheng Chao, Wang He, Yang Jing, Li Hongjia, and Wang Liming. Fedshe: privacy preserving and efficient federated learning with adaptive segmented ckks homomorphic encryption. Cybersecurity, 7(1): 40, 2024.
>
> [2] Neveen Mohammad Hijazi, Moayad Aloqaily, Mohsen Guizani, Bassem Ouni, and Fakhri Karray. Secure federated learning with fully homomorphic encryption for iot communications. IEEE Internet of Things Journal, 11(3):4289–4300, 2024. doi: 10.1109/JIOT.2023.3302065.
>
> [3] Siddhant Dutta, Pavana P Karanth, Pedro Maciel Xavier, Iago Leal de Freitas, Nouhaila Innan, Sadok Ben Yahia, Muhammad Shafique, and David E. Bernal Neira. Federated learning with quantum computing and fully homomorphic encryption: A novel computing paradigm shift in privacy-preserving ml, 2024. URL https://arxiv.org/abs/2409.11430.
>
> [4] Siddhant Dutta, Nouhaila Innan, Sadok Ben Yahia, Muhammad Shafique, and David Esteban Bernal Neira. Mqfl-fhe: Multimodal quantum federated learning framework with fully homomorphic encryption, 2025. URL https://arxiv.org/abs/2412.01858.

---

### Review · Reviewer_mJTq · 2026-01-30

**Summary Of Contributions:**

This paper proposes an end-to-end Quantum Fеderated Learning (QFL) pipeline where clients еncrypt‍ their model updates using CKKS-based fullу homomorphic encryption (FHE), and then utilizеs that implementation to measure the practical сosts. The primary contribution is a clear, emрirical breakdown‍ of the systems overhead intrоduced by FHE in this setting, reporting (i) trаining/round-time components (including encryptіon/decryption and server-side aggregation), (іi) client and central-party memory/CPU usage, аnd (iii) per-round communication volume, across multіple model variants (classical CNN/ResNet baselіnes and hybrid QNN/QCNN variants). Overall, thе results suggest that encryption/decryption itself can be relatively inexpensive, but the servеr-side aggregation, memory usage, and especially communication costs grow dramatically under FНE, making naive "encrypt everything" deployment‍ difficult without further optimization.

**Additional Comments:**

I am relatively new to quantum machine learning and not deeply specialized in homomorphic encryption, so I may be missing relevant prior work or subtle nuances. My comments focus on what is directly evidenced in the submission and on the clarity and reproducibility improvements that would help a broad ML audience.

**Audience:**

Yes

**Audience Explanation:**

Yes, readers interested in privacy-preserving ML systems, federated learning, and practical deployment costs of cryptographic methods would likely find the quantitative overhead results useful.

**Broader Impact Concerns:**

No major broader impact concerns beyond standard considerations for privacy-preserving ML.

**Claims And Evidence:**

Yes

**Claims Explanation:**

The core claims made, which are about the resource overhead, are supported by evidence in the Results section. Measurements across multiple model variants are reported.

That being said, I think some additional experiments ablating over the number of clients, the number of parameters encrypted, etc, would help improve the generalizability of the claim.

**Requested Changes:**

1. The paper should specify all experimental and implementational details to ensure reproducibility. For example, the "CNN" model architecture used is not described (number of layers, channels, etc).

2. The configuration parameters used in CKKS/TenSEAL must also be outlined. I am not too well-versed in this topic, but I believe this would also influence the overheads. So additional ablations showing this trade-off would be beneficial too.

3. From a strictly federated learning perspective, ablations on the number of clients, participation rate, etc would be needed. If full sweeps are computationally infeasible, even a limited subset (e.g., 5/10/20 clients) or a discussion of expected scaling based on measured components would be useful.

4. The paper should clarify how the communication cost is computed. Is it an end-to-end measurement or is it estimated from parameters?

---

> ### Author Response · Authors · 2026-02-04
> **Authors' Response to Reviewer mJTq (I)**
>
> # Response
> The authors thank the reviewer for the positive and constructive feedback as well as for highlighting important aspects of reproducibility and experimental transparency.
>
> ## Model Architecture
> >The paper should specify all experimental and implementational details to ensure reproducibility. For example, the "CNN" model architecture used is not described (number of layers, channels, etc).
>
> To address the concern about missing architectural details, we are providing full transparency by making the complete source code and comprehensive documentation publicly available, which will enable exact replication of all experiments. To make the aforementioned details more accessible without browsing the source code, we added an appendix section containing the relevant information.
>
> ## Configuration Parameters
> >The configuration parameters used in CKKS/TenSEAL must also be outlined. I am not too well-versed in this topic, but I believe this would also influence the overheads. So additional ablations showing this trade-off would be beneficial too.
>
> Regarding the CKKS/TenSEAL configuration, we agree that these parameters are relevant and impact memory and communication overhead. Recent work has shown that CKKS parameters can be optimized for a given security level, directly affecting performance and overhead [1]. To make these implementation choices explicit, we added this information to the appendix as well. The same applies to the number of clients, the participation rates and the way the metrics are computed.
>
> # Summary of Revisions
> - Implementation Details: Added a detailed list of collected metrics and how they are derived, e.g., communication overhead. See Chapter A.1, p. 17.
> - Experimental Configuration: Provided full configuration of our experiments, including number of clients and CKKS parameters. See Chapter A.2, p. 17.
> - Model Architecture Details: Added tables in the appendix listing each layer and corresponding operations of the underlying models. See Chapter A.3, p. 17.
>
> # References
> [1] Yao Pan, Zheng Chao, Wang He, Yang Jing, Li Hongjia, and Wang Liming. Fedshe: privacy preserving and efficient federated learning with adaptive segmented ckks homomorphic encryption. Cybersecurity, 7(1): 40, 2024.

---

### Author Response · Authors · 2026-02-04
**Revision Summary**

Dear Reviewers,

We would like to express our sincere gratitude for the thoughtful and constructive feedback provided during the review process. In response, we have carefully revised our manuscript to address the comments and enhance the overall quality of our work. Below is a summary of the all changes we have made since the last submission:

- **Implementation Details:** Added a detailed list of collected metrics and how they are derived, e.g., communication overhead. See Chapter A.1, p. 17. (Requested by reviewer mJTq)
- **Experimental Configuration:** Provided full configuration of our experiments, including number of clients and CKKS parameters. See Chapter A.2, p. 17. (Requested by reviewer mJTq)
- **Model Architecture Details:** Added tables in the appendix listing each layer and corresponding operations of the underlying models. See Chapter A.3, p. 17. (Requested by reviewer mJTq)
- **VQC Parameter Updates:** Clarified that parameters of our VQCs are iteratively updated using conventional machine learning methods on a classical computer, as our experiments are simulations. See Chapter 3.1, p. 3. (Requested by reviewer 5peD)
- **Quantum Runtime Statements:** Corrected statements regarding runtimes of quantum simulations versus actual quantum computers with small numbers of qubits. See Chapter 5.1, p. 7. (Requested by reviewer 5peD)
- **Quantum Hardware Limitations:** Added a note that our study does not capture overhead from real quantum hardware, e.g., gradient estimation or state collapse upon measurement. See Chapter 6.2, p. 12. (Requested by reviewer 5peD)
- **Gradient Aggregation:** Clarified that FedAvg was used to aggregate model updates. See Chapter 4, p. 5. (Requested by reviewer Dspu)
- **Related Work Update:** Included Jiang et al. (2025) in the related works section. See Chapter 2, p. 2. (Requested by reviewer Dspu)

We hope these changes effectively address your concerns and further strengthen our submission. Thank you once again for your valuable feedback and time.

We are available to answer any further questions.

Sincerely, TMLR Paper6569 Authors

---

### Decision · Action_Editor_8di9 · 2026-03-10

**Recommendation:** Reject

**Audience:**

No

**Audience Explanation:**

While privacy in federated learning is a timely topic, the findings lack practical relevance for both the classical and quantum machine learning communities. The results merely confirm the well-known massive overhead of naive FHE without exploring modern cryptographic optimizations, and the chosen quantum models significantly underperform classical baselines on the selected tasks. Consequently, the simulated environment is too detached from realistic quantum constraints to provide actionable insights for the TMLR audience.

**Claims And Evidence:**

No

**Claims Explanation:**

The claims regarding the resource overhead of Quantum Federated Learning are unsupported because the experimental setup relies entirely on classical CPU simulations, ignoring the massive constraints of actual quantum hardware. Specifically, it fails to account for the severe computational costs of estimating quantum gradients (e.g., via the parameter-shift rule), which would likely dwarf the reported encryption overhead. Furthermore, the omission of critical experimental details, such as FHE configurations and specific model architectures, hinders reproducibility.